# Improving the Customs Regulation Framework in the Eurasian Economic Union in the Context of Sustainable Economic Development

Natalia Vovchenko [1,*], Olga Ivanova [1], Elena Kostoglodova [1], Stanislav Khapilin [1] and Karina Sapegina [2]

[1]  Faculty of Economics and Finance, Rostov State University of Economics, 344002 Rostov-on-Don, Russia; sovet2-1@rsue.ru (O.I.); ramachka2006@rambler.ru (E.K.); khapilin@mail.ru (S.K.)
[2]  Institute of Industrial Management of Economics and Trade, Peter the Great St. Petersburg Polytechnic University, 195251 Saint-Petersburg, Russia; sapegina.k@edu.spbstu.ru
[*]  Correspondence: nat.vovchenko@gmail.com

**Abstract:** The formation of a customs administration framework based on the digital economy in the Eurasian Economic Union (EAEU) requires the application of fundamentally new technologies. The successful implementation of digital technologies in the information space of the EAEU presupposes the solution of a number of problems associated with the ensuring the implementation of the concept of sustainable development of the EEU member states in the new economic reality and transition to a new paradigm of customs administration based on the digitalization of the processes of regulation of foreign economic activity. Based on this paradigm, we set the following tasks: to identify trends and substantiate the need for digitalization of the customs administration mechanism in the Eurasian Economic Union based on the use of new technologies; to reveal the meaningful features of digital technologies that are promising for the development of the mechanism of customs administration of the EAEU; consider the applied aspects of the latest information technologies used in the course of EAEU customs administration system digitalization; and assess the prospects for their use, analyze the prospects of organizational, legal and managerial support of this process in the EAEU at the supranational and national levels. The article concludes that within the framework of the digital transformation of the EAEU, new opportunities are opening up for the customs regulation framework, based on the introduction of technologies for analyzing large amounts of data, immersive technologies, blockchain, the use of innovative methods for obtaining and processing customs information (satellite tracking, radio frequency identification), and the introduction of artificial intelligence technologies in customs control processes.

**Keywords:** big data; Eurasian Economic Union; customs administration; digital transformation; G28; G38; F02; F15



## 1. Introduction

The recent years of the development of Eurasian economic integration have been characterized by contradictory trends. On the one hand, the EAEU is gradually removing obstacles in the domestic market and liberalizing the service market, launching single markets for medicines and medical devices, forming an electricity market, establishing transparent rules for regulating the labor market, and implementing uniform measures to protect the common market and control cross-border markets. On the other hand, obvious organizational and regulatory achievements are accompanied by dithering of the economic basis of the union, a decrease in the rate of economic cooperation, insignificant achievements in the real sector of the economy and scientific and technical cooperation, which is of fundamental importance for the progressive provision of economic growth and the realization of the national interests of the EAEU member states.

The coronavirus pandemic has shaken the global economic model and created risks for long-term economic growth associated with reduced demand, reduction and closure of production capacity, deterioration in the quality of human capital, disruption of production and technological chains, identified the most vulnerable spots of national economies, and became a strength test for most integration associations of the world. The introduction of restrictive measures in many countries of the world due to the coronavirus pandemic and the economic crisis predetermined the deterioration of the stability of the economic systems of the EAEU countries, despite the fact that the growth of the EAEU GDP over the past 5 years was inferior to global economic growth.

According to the estimates of the Eurasian Economic Commission, the GDP of the EAEU countries in 2020 decreased by 3.0%. The reduction is spotted in all EAEU member states (Figure 1).

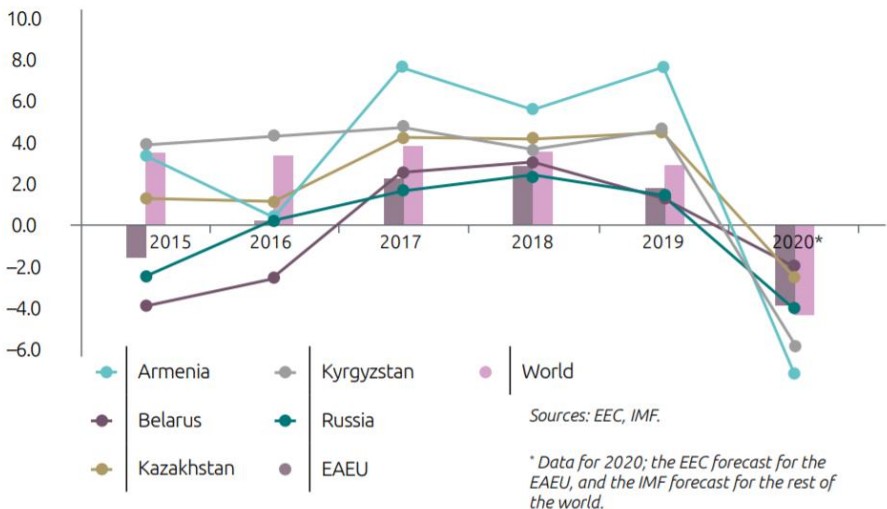

**Figure 1.** GDP growth rates of the EAEU member states and around the globe, percentage [1].

Negative external factors also affected other indicators of economic activity in the EAEU region. At the end of 2020, the industrial production index compared to 2019 amounted to 97.6%, the volume of mutual trade—89.3%, the volume of investments decreased by 1.9% (Figure 2).

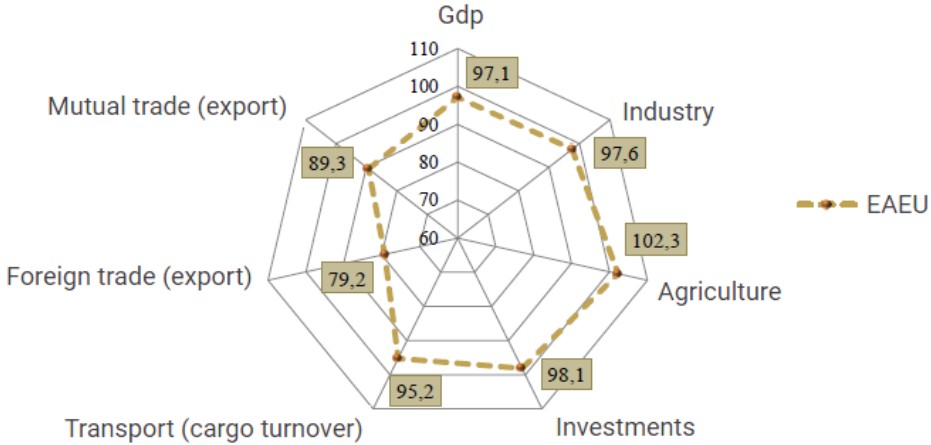

**Figure 2.** Main economic indicators of the EAEU (in percent by 2019, in comparable prices) [2].

The trade turnover of the EAEU member states with third countries in 2020 decreased by 20.9%. The growth of exports to third countries in 2020 was noted only in the Kyrgyz Republic—105.5% (up to USD 1.4 billion). In the rest of the EAEU member states, there was a decrease in the value of exports: in the Russian Federation, the export growth index

amounted to 78.6% (USD 304.7 billion), in the Republic of Belarus—81.8% (USD 15 billion), in the Republic of Kazakhstan—80.2 (USD 41.4 billion), and in the Republic of Armenia—97.7% (USD 1.8 billion).

In these conditions, it is necessary to develop flexible mechanisms for targeted assistance to economic development, aimed at the implementation of joint cooperation projects and their financial support on the basis of the EAEU institutions, and the creation and development of highly productive sectors of the economy. This requires the completion of the formation of a single economic space based on digital transformation and a system of strategic and indicative planning, the development of conceptual approaches to the formation of mechanisms for equalizing the levels of economic development of the EAEU member states and ensuring sustainable economic growth, and determination of the list of priority integration infrastructure projects.

## 2. Methodology and Related Previous Work

The scientific hypothesis of the study is the assumption that, in the context of the spread of the coronavirus pandemic, there are significant changes in the economy and new challenges, expressed in such trends as: the presence of a high dependence of economic development on the epidemiological situation, uneven recovery of business activity between countries, restraint of inclusive recovery of the world economy, a decrease in the quality of human capital, etc. [3] All this affects the achievement of the UN sustainable development goals until 2030 and requires a change in the paradigm for the development of Eurasian economic integration, attracting additional sources of economic growth associated with the widespread use of digital platforms, and on this basis revising conceptual approaches to the system of customs administration in the EAEU member states.

Currently, there is a wide range of views in the economic literature regarding the directions of development of Eurasian economic integration and ensuring sustainable economic growth, the role of digitalization in this process.

In recent years, a large number of studies have appeared on the strategic directions of the EAEU development, the transformation of economic models in the countries of the post-socialist world [4,5], the conjugation of the Eurasian and other integration projects [6–10], key areas of modernization of economic cooperation [11–13], the development of EAEU transport corridors [14], and the formation of a unified monetary and financial policy [15].

The digital agenda has recently been implemented within the EAEU. The main strategic document was adopted in 2017; therefore, this area of research is actually at the stage of formation. In general, research on the impact of the digital agenda on the economic development of the EAEU member states, as an independent direction in the academic field, is in its infancy. Moreover, there are not many works devoted to a comprehensive assessment of the digital agenda or its place in general integration processes. This is due to the fact that the agenda has been actively implemented for only three years; not much time has passed in order to conduct a comprehensive analysis of the results or problems. However, such studies have already been conducted [16–18], analyzing the implementation of national digitalization programs, and their relationship with the Eurasian agenda [19–22].

A significant number of publications are available on more specific aspects of the agenda related to its practical implementation. Thus, special attention to the processes related to the digital agenda is paid by researchers in the field of foreign trade regulation, since the most active digitalization processes in the EAEU are taking place in electronic commerce. To date, the possibilities of forming digital transport corridors and developing on this basis the transit potential of the EAEU, regulation of cross-border electronic document flow [23,24] are being widely considered, opportunities for the protection of intellectual property rights in the EAEU are analyzed as part of the implementation of the digital agenda, and risks in this area [25].

The studies carried out combine the following key theses on the high role of digital technologies as an additional source of economic growth and ensuring economic stability, as well as the EAEU's potential for the successful development of digital infrastructure.

At the same time, in the scientific studies available today, insufficient attention is paid to the customs component of digitalization processes in the EAEU, which determines the need for such a study and consideration of the following tasks: researching the substantive characteristics of digital technologies promising for the development of the customs administration mechanism, considering their applied aspects, and assessing prospects for use in the Eurasian Economic Union at the supranational and national levels.

To solve the set tasks, the following methods were used in the work: abstract-logical and theoretical generalization in order to substantiate the substantive characteristics of the process in the field of Eurasian integration and improve customs administration; system-integrated and risk-oriented approaches in the face of new global challenges and economic uncertainty, which implies the search for effective methods of risk identification, the use of risk management tools and stress testing as part of digital transformation.

### 3. Results

The intensive development of integration processes in the world economy is taking place in close relationship with digitalization processes entering into all areas of economic activity of states, which in turn has a strong impact on the transformation of the structure and nature of mutual relations between countries in the field of trade and production. New sectors of the economy are being formed, innovations are being introduced everywhere, transaction costs are decreasing, which allows business structures to gain additional competitive advantages, and, within the framework of integration processes, to increase the speed of trade and expand opportunities for intra and inter-sectoral cooperation. As a result, most regional integration associations today include digitalization in their strategic documents as the main trend of future development. The Eurasian Economic Union is no exception. Within the EAEU, significant progress is currently being noted in many areas of digital development. There are a number of prerequisites for the development of such a process, among which we highlight, first of all, the capacity of information technology in the EAEU countries. This is evidenced by a number of indicators, such as the digital technology adoption (DIT) index, as well as known rating indicators—the UN e-government development index, and the information and communication technologies (ICT) development index. In the ratings presented by the UN among 193 countries of the world regarding the state of the level of development of electronic government according to 2020, the EAEU countries were also ranked quite high: Kazakhstan was ranked 29th, Russia—36th, Belarus, Armenia, and Kyrgyzstan were 40th, 68th, and 83rd., respectively. According to ICT development index in the countries in the ranking of 176 countries of the world for 2017, Belarus, Russia, and Kazakhstan were 32nd, 45th, and 52nd, respectively; Armenia—75th; and Kyrgyzstan only 109th. Russia and Kazakhstan are undoubtedly leaders in the field introduction and distribution of digital technologies and services in the EAEU region.

The prospects and relevance of the digital development of the economies of the EAEU states are reflected in the main areas of implementation of the EAEU Digital Agenda until 2025. The document was adopted at the level of the Eurasian Intergovernmental Council to study the implementation of the Main Directions—the Procedure for developing initiatives within the framework of the EAEU digital agenda (Table 1).

The Eurasian Economic Commission formulated the principles of digital modernization of economic processes, which should determine the grounds for the formation of a set of projects and rule-making initiatives in this area:

1. The use of an architectural approach when describing processes within the Union and agreeing on common processes, projects, and subprojects implemented on the basis of aggregation of common processes, taking into account all components (glossary, normative and reference information, catalogs, registers, and mechanisms).
2. Creation of a digital platform based on the integrated information system of the EAEU, which should provide a comprehensive set of reference entities to ensure the

    functioning and development of digital service platforms in existing and emerging industries and industries.

3. Introduction of a system for assessing risks and effectiveness of current and future general processes, architecture changes.
4. Introduction of a unified methodology for the development of integration processes and their components: a unified methodology, requirements, regulations, including harmonization at the level of EAEU regulatory legal acts, at the level of the member states (for deep seamless integration, real national plans for digital integration and transformation, unification of regulatory legal acts, reengineering, organizational transformations, training, implementation and technical support, and life cycle planning process).

The goal of implementing the digital agenda is determined by:

- Formation of new directions of integration cooperation, taking into account the digitalization processes in the world economy;
- Ensuring high-quality and sustainable economic growth of the EAEU member states, including in order to accelerate the transition to a new technological order and creating conditions for the industries of the future on the territory of the EAEU.

**Table 1.** Digital space of the EAEU until 2025 [26].

| Digital Space Element | Component of an Element |
|---|---|
| Digital solutions | Electronic customs<br>Electronic commerce<br>Electronic logistics<br>Digital finance/fintech<br>EAEU digital integration platform |
| Digital infrastructure | Electronic identification services<br>Information Security<br>Cloud infrastructure and initiatives |

    The economic effect from the implementation of the digital agenda for the future until 2025 is estimated at 10.6% of the total expected growth in the total GDP of the EAEU member states by 2025. Only in the service sector, by removing barriers, it is possible to save about USD 50 billion, which will significantly increase the volume of export of IT services. Moreover, the development of digitalization will contribute to the activation of migration processes, the creation of jobs, affect the increase in labor productivity, quality of services and, in general, the competitiveness of the Eurasian Union of the EAEU. According to expert estimates, the EAEU countries will receive the so-called "digital dividends" consisting in increasing economic growth, migration activity and expanding the working space, improving the quality of services in various areas, as well as increasing the competitiveness of the Eurasian region as a whole [27].

    As part of the implementation of a single digital agenda, the EAEU member states have formulated the goals and objectives of digitalization in the field of customs regulation of foreign economic activity, mainly in the form of strategic documents for the development of national customs services. The key tasks of the customs component of the digital agendas of the EAEU member states are the introduction of promising customs technologies (electronic transit, automatic registration, and release of goods), the introduction of a system of mandatory labeling of goods. The differences lie in the objects of digital transformation, depending on the structure of customs authorities, technical equipment of checkpoints, the state of development of national single window systems, and electronic interdepartmental interaction [28].

    In addition, the guidelines for digitalization are determined by the EAEU Customs Code, which came into force in 2018. The provisions of the code are based on the initiatives of the World Customs Organization in the field of information technology and provide

for the transition to electronic document management in the customs sphere, mandatory preliminary electronic notification of the arrival of goods in the customs territory of the EAEU, normatively fix the possibility of implementation by national customs authorities of technologies for automatic registration and automatic release of goods, and introduction of single window mechanisms.

With regard to the introduction of electronic declaration in Russia, a new framework has been built in the period from 2018 to 2020. The number of customs clearance points has been significantly reduced—to 16 electronic declaration centers instead of more than 600 customs posts that previously processed declarations. As a result of the reform, the Federal Customs Service created electronic customs offices and subordinate electronic declaration centers (EDCs) in all federal districts; four specialized EDCs by types of transport (Aviation, Baltic, Novorossiysk, and Vladivostok); two specialized EDCs by types of goods (Excise and Energy); and two territorial EDCs (Kaliningrad and Moscow regional).

Algorithms for automatic registration and issuance of declarations for goods, which are currently used by the customs authorities of the EAEU member states, are implemented directly in the code of the software used to register goods. At the same time, the architectural solutions built into them do not allow automating some of the technological features of declaring goods, in particular, ensuring the automatic registration of preliminary customs declarations. In this regard, a new approach is needed, unified at the level of the Eurasian Economic Commission and based on the conduct by customs officials only of those checks on which the information system of the customs authorities cannot make a decision on their own.

This requires the improvement of algorithms for automatic and automated decision-making and regulatory and reference information, in particular:

- Format and logical control, eliminating the introduction of declarations for goods filled in with errors;
- Algorithms for automatic distribution of declarations between customs posts;
- Algorithms for automatic and automated decision-making in certain areas of checking goods declarations (checking the correctness of classification, country of origin, customs value of goods, completeness of payment of customs duties, etc.);
- Classifiers, reference, and registry information (in terms of ensuring the possibility of using methods of semantic analysis of unstructured text data;
- Creation of software tools that ensure traceability of electronic documents along the entire technological chain of customs clearance of goods, as well as one-time provision of information for all types of state control;
- Ensuring fault tolerance of the software, since when the speed of information processing slows down, technical failures become the reasons for refusing to automatically perform customs operations and their commission by customs officials.

One of the priority areas of digitalization of customs administration in the EAEU is the creation of "single window" systems to ensure the submission of information to authorized bodies and the coordinated development of electronic forms of interaction between customs, other regulatory bodies and participants in foreign economic activity. Thus, in the Republic of Kazakhstan, foreign trade participants receive a significant number of permits in electronic form using the state database "Electronic Licensing". In the Republic of Kyrgyzstan, the Center for Foreign Trade "Single Window" was created, the information system "Tulpar System" was put into operation, which provides electronic submission of applications by participants in foreign economic activity to state bodies for obtaining permits. A system of interdepartmental electronic interaction has been introduced and is actively being developed in Russia

At the same time, the disadvantages of the implemented systems are poor coordination of participants, lack of communication of the technologies used with international forms, and standards for the presentation of information. In electronic form, requirements for the provision of paper documents along with electronic ones, weak interdepartmental interaction, low interest of participants in foreign economic activity, etc. [29,30].

The need to intensify work to ensure effective information interaction in the EAEU led to the approval by the decision of the Eurasian Intergovernmental Council of 30 April 2019 No. 6 of the reference model of the national single window mechanism in the system of regulation of foreign economic activity, which involves the creation of an ecosystem for managing foreign economic activity by integrating all data elements from various sources and data exchange between business entities, state bodies of the EAEU member states and authorized organizations. The architecture of the reference model provides for the interface of information systems of state bodies of the EAEU member states and authorized organizations, including customs, with the integration gateway of the EAEU Unified Information System through the systems of interdepartmental electronic interaction.

The model includes extensive functionality of national single window systems:

- For state bodies—performance of state functions and provision of state services in the field of regulation and control of foreign economic activity (including coordinated management at the customs border) and tax regulation, use of an interdepartmental risk management system, use of data, control results, and decisions of state bodies of the Member States EAEU and third countries;
- For foreign trade participants—digitalization of providing the necessary data, performing other legally significant actions, obtaining decisions of state bodies of the Member States and authorized organizations, the ability to manage the processes of transportation, warehousing, storage of goods within the supply chain, interaction with banks and other financial organizations for financial management, the possibility of using the "Personal Account" service with the functionality of collecting information, and reporting, archiving, and reusing data [31,32].

The promising digital technologies in the field of customs administration also include technologies for analyzing big data and using artificial intelligence in customs control processes. Big data include significant amounts of information with a fairly large variety of them that could be accumulated and processed by software and become the basis for introducing elements of artificial intelligence into the planning of the activities of customs authorities, registration and issuance of customs declarations, identification of objects of customs control with an increased risk of violation of customs legislation.

In many cases, the interpretation of big data (including partially structured data) requires the use of artificial intelligence and neural network technologies, which will make possible adapting to changing conditions as quickly as possible, a universal approximation of a function of several variables, as well as resistance to errors due to a variety of interneural connections. The applied nature of these technologies is most evident in the following areas:

- Formation and subsequent analysis of customs statistics of foreign and mutual trade of the EAEU states;
- Forecasting budget revenues from the receipt of customs and other payments administered by customs authorities;
- Analysis of inspection equipment images;
- Revealing the facts of inaccurate classification of goods based on the analysis of its characteristics;
- Comparison of declared goods with various restrictive lists and registries.

Real digitalization of the sphere of customs administration of the EAEU is impossible without the formation of a unified system for the exchange of digital data based on the technological compatibility of the digital solutions and platforms used. The basis for the implementation of these processes on the territory of the EAEU should be the formation of a single space of electronic trust, ensuring the safety and reliability of interstate exchange of data and electronic documents. The Strategy for the development of a cross-border space of trust approved by the EEC in the long term until 2024 provides for the provision of the actual possibility of electronic interaction of individuals and legal entities located in different states of the Union with each other, as well as with public authorities. It should be noted that the implementation of the cross-border space of electronic trust, which makes it

possible to recognize digital signatures and organize electronic document flow within the EAEU, to provide access for legal entities and individuals to legally significant customs services both within the EAEU and in third countries, requires a fundamental solution of the following questions:

− formation of a basic model of threats to the security of information and actions of violators in data transmission channels between the integrated system and information systems of international associations and third countries;
− development of a concept for the use of legally significant electronic documents and services in the interstate information interaction of legal entities of the EAEU member states with each other and with authorized state bodies;
− development of uniform standards for the formation of electronic documents created in different countries, and ensuring their compatibility;
− creation of a system for identification and authentication of subjects of electronic interaction with a high degree of reliability;
− ensuring the protection of electronic documents, taking into account the requirements of the legislation of the EAEU member states and international treaties.

The formation of an efficiently functioning EAEU customs administration framework presupposes the need to unify the approaches of the member states to the modernization of checkpoints based on the principles of the digital agenda. Since 2020, the Federal Customs Service has begun to implement the concept of an intelligent automobile checkpoint. The architecture of the model consists of mutually integrated information and technological elements, optimally built and adapted to the conditions of each checkpoint based on the existing infrastructure, geographic and social conditions, which will create conditions for continuous "risk-free" supplies through checkpoints at those sections (stages) where solutions are implemented that automate the state control process without losing its quality. At the same time, the disparate approaches of the EAEU member states to the arrangement of checkpoints may ultimately lead to a rupture of the "external contour" of the system of customs regulation of foreign economic activity in the EAEU. The objective of increasing the efficiency of the EAEU customs regulation system requires a clear consolidation of standards for maximum automation of customs operations using elements of artificial intelligence, which should be used by all EAEU member states during the reconstruction, modernization and construction of checkpoints.

Blockchain technology, which underlies cryptocurrencies and decentralized information exchange systems, is also important for solving problems and tasks in customs administration system.

The greatest potential of blockchain technology is predicted in the following areas:

• Sharing by customs authorities of information about the supply chain of participants in foreign economic activity to other regulatory authorities through the blockchain, in particular, tax authorities for tax control purposes;
• Programming of the goods tracking system similar to the technology of smart contracts;
• Comprehensive analysis of product supply chains to improve the risk management system;
• Verification of certificates of origin of goods and control of the country of origin of goods.

## 4. Conclusions and Recommendations

In the Eurasian Economic Union, which faces the global task of shaping the economy of the future, the transition to the digital economy is seen as a key factor in economic growth and ensuring economic stability. In recent years, the EAEU has made significant progress in many areas of digital development. The field of customs administration is at the forefront of digital transformation, using the latest technologies in machine learning, robotics and artificial intelligence. The adoption of the EAEU Customs Code laid the legal basis for activating digitalization processes in the customs sphere, the most important areas of which were: interaction of customs authorities with foreign economic activity

participants mainly in electronic form, automation of customs operations, introduction of "one window" technologies when moving goods across the customs border and interaction with other regulatory authorities.

The future of the digital breakthrough depends on the ability of regulators to create favorable conditions for further digital transformation in the EAEU. The presented results of the analysis of trends in the digital development of the EAEU countries in the field of customs administration indicate that the new joint digital strategy should use a wider set of mechanisms than those that are currently included in the sphere of digital policy, namely, it is necessary:

−   Formation of digital trading platforms on the basis of "one window" systems, ensuring the implementation of the concept of a continuous trading process, achieving "seamlessness" of economic processes and service environment.
−   Introduction of artificial intelligence technologies into the processes of performing customs operations during the customs declaration of goods and the movement of goods and vehicles at checkpoints across the customs border of the EAEU;
−   Expanding the use of big data for the purposes of developing customs policy, generating customs statistics, administering income, and auditing performance;
−   Formation of the regulatory and legal framework of the Eurasian Economic Union and the member states, necessary for the implementation of interstate electronic services and processes of cross-border exchange of legally binding electronic documents.

**Author Contributions:** Conceptualization, N.V. and O.I.; methodology, E.K. and S.K.; data curation, E.K and S.K.; writing—original draft preparation, N.V., K.S., O.I. and E.K.; writing—review and editing, O.I., K.S. and E.K.; project administration, N.V.; funding acquisition, N.V. All authors have read and agreed to the published version of the manuscript.

**Funding:** The research is partially funded by the Ministry of Science and Higher Education of the Russian Federation under the strategic academic leadership program 'Priority 2030' (Agreement 075-15-2021-1333 dated 30 September 2021).

**Institutional Review Board Statement:** Not applicable.

**Informed Consent Statement:** Not applicable.

**Conflicts of Interest:** The authors declare no conflict of interest.

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
