# Peer review of "Improving the Customs Regulation Framework in the Eurasian Economic Union in the Context of Sustainable Economic Development"

_sustainability, doi:10.3390/su14020755_

Round 1

Reviewer 1 Report

I would suggest authors clarify the content of digital agenda they refer to in the article. I do not feel the sufficiency of citations the authors rely on and improvement can be made to the citations.

Author Response

Good day! 

Thanks for yor review

Changes made to the article in accordance with the comments of the reviewers:

1) section 2 "Methodology and Related Previous Work" sets out the hypothesis, plan and research methods;

2) supplemented and clarified the theses concerning the improvement of technologies for the automatic release of goods, the creation of “single window” systems, the formation of a single space of electronic trust in the EAEU, as well as the section “Conclusions and recommendations”.

3) the list of references increased by 23%;

4) the content of the EAEU digital agenda until 2025 is disclosed;

5) added links to the work of foreign researchers on the subject of the article;

6) 1 figure and 1 table have been added, as well as clarifications and additions have been made to section 4 “Conclusions and recommendations”;

7) added statistical data, including in a graphical form, on the dynamics of the economic development of the EAEU in the context of substantiating the directions for the formation of a single economic space based on the digital transformation of the sphere of customs administration;

8) section 2 "Methodology and Related Previous Work"  describes the contribution of research to academic development.

Reviewer 2 Report

I think the topic would be interesting and important in the near future. I suggest the authors to improve the reference list which contains only 26 items. It should be necessary to involve more international literature sources which increase the level of the paper. In my mind the paper is descriptive, so it would be better to find the way of how is it possible to change it. Probably you can present more tables and figures which are related the topic and help the understanding. I suggest to improve the 4th chapter too.

Author Response

Good Day!

Thanks for your review

Changes made to the article in accordance with the comments of the reviewers:

1) section 2 "Methodology and Related Previous Work" sets out the hypothesis, plan and research methods;

2) supplemented and clarified the theses concerning the improvement of technologies for the automatic release of goods, the creation of “single window” systems, the formation of a single space of electronic trust in the EAEU, as well as the section “Conclusions and recommendations”.

3) the list of references increased by 23%;

4) the content of the EAEU digital agenda until 2025 is disclosed;

5) added links to the work of foreign researchers on the subject of the article;

6) 1 figure and 1 table have been added, as well as clarifications and additions have been made to section 4 “Conclusions and recommendations”;

7) added statistical data, including in a graphical form, on the dynamics of the economic development of the EAEU in the context of substantiating the directions for the formation of a single economic space based on the digital transformation of the sphere of customs administration;

8) section 2 "Methodology and Related Previous Work"  describes the contribution of research to academic development.

Reviewer 3 Report

Referee Report on “Improving the customs regulation framework in the Eurasian Economic Union in the context of sustainable economic development”_ sustainability-1503214

This study based on the successful implementation of digital technologies in the information space of the EAEU presupposes the solution of a number of problems associated with the transition to a new paradigm of customs administration based on the digitalization of the processes of regulation of foreign economic activity. The author concludes that within the framework of the digital transformation of the EAEU, new opportunities are opening up for the customs regulation framework, based on the introduction of technologies for analyzing large amounts of data, immersive technologies, blockchain, the use of innovative methods for obtaining and processing customs information (satellite tracking, radio frequency identification), the introduction of artificial intelligence technologies in customs control processes.

However, this research neither proposes economic reality analysis and argumentation, nor does it provide theoretical support. This journal is an academic journal; therefore, it is not suitable for publication in this journal in current situation. I suggest that the author should appropriately use words, numbers, tables, and diagrams to construct his theoretical framework, describe and explain the findings, as well as the related problems currently faced, and propose solutions.

It is recommended that the author explain the position and contribution of this research in academic development.

Minor Comments:

 The authors should carefully check the symbols, spelling and writing.

  1. For example, in line 119 and 122, the first letter of the sentence should be capitalized the sentence. In line 147, “fourteen” should be “14”.

Evaluation:

For the above reasons, I believe that the current situation in this article is not suitable for publication in this journal. I encourage authors to make submissions after making appropriate corrections.

Author Response

(The authors gave the same response as above.)

Round 2

Reviewer 2 Report

Thank you for your work and improvement of the paper, I appreciated it.

Reviewer 3 Report

Thanks to the authors' efforts, I think the article can be published after correction.